

# A Dataset for Investigating Socio-ecological Changes in Arctic Fjords

Robert W. Schlegel[1], Jean-Pierre Gattuso[1,2]

[1]Laboratoire d'Océanographie de Villefranche, Sorbonne University, CNRS, Villefranche-sur-mer, France
[2]Institute for Sustainable Development and International Relations (IDDRI-Sciences Po), Paris, France

*Correspondence to*: Robert W. Schlegel (robert.schlegel@imev-mer.fr)

**Abstract.** The collection of *in situ* data is generally a costly process, with the Arctic being no exception. Indeed, there has been a perception that the Arctic lacks for *in situ* sampling; however, after many years of concerted effort and international collaboration, the Arctic is now rather well sampled with many cruise expeditions every year. For example, the GLODAP
product has a greater density of *in situ* sample points within the Arctic than along the equator. While this is useful for open ocean processes, the fjords of the Arctic, which serve as crucially important intersections of terrestrial, coastal, and marine processes, are sampled in a much more *ad hoc* process. This is not to say they are not well sampled, but rather that the data are more difficult to source and combine for further analysis. It was therefore noted that the fjords of the Arctic are lacking in FAIR (Findability, Accessibility, Interoperability, and Reuse) data. To address this issue a single dataset has been created
from publicly available, predominantly *in situ* data from 7 study sites in Svalbard and Greenland. After finding and accessing the data from a number of online platforms, they were amalgamated into a single project-wide standard, ensuring their interoperability. The dataset was then uploaded to PANGAEA so that it itself can be findable and reusable into the future. The focus of the data collection was driven by the key drivers of change in Arctic fjords identified in a companion review paper. To demonstrate the usability of this dataset an analysis of the relationship between the different drivers was
performed. Via the use of an Arctic biogeochemical model these relationships were projected forward to 2100 via RCP 2.6, 4.5, and 8.5. This dataset is a work in progress and as new datasets containing the relevant key drivers are released they will be added to an updated version planned for mid 2024.

The dataset (Schlegel & Gattuso, 2022) is publicly available on Zenodo at: https://doi.org/10.5281/zenodo.7472376

## 1 Introduction

The Arctic is a region of extreme contrasts. In the winter, life must contend with constant darkness, sea water that can freeze solid, and a pervasive silence punctuated only by violent gusts of wind. Whereas the summer has 24 h of daylight, dramatically warmer air temperatures, and the arrival of migratory seabirds for the noisey business of breeding (Deschamps et al., 2019). While much of the Arctic tundra is relatively barren throughout the year compared to ecoregions further south, coastal Arctic waters can be teaming with life. Of these systems, fjords tend to provide the most diverse range of habitats for



many important species. Thereby acting both as sources for extractive human activities, as well as possible refuges against some of the oncoming ravages of climate change (Węsławski et al., 2011; Bonnet-Lebrun et al., 2022).

The rate of loss for the Arctic cryosphere driven by the changing climate is alarmingly rapid (Schlegel et al., in review), making it critical that Arctic fjord systems be as actively monitored as possible. Even though this monitoring in the extreme
north is both challenging and costly, there has been a concerted international effort to maintain and increase it. The majority of the resulting *in situ* data collection is however done via large cruise ships, and the sampling of data throughout all but the mouths of fjords tends to be limited due to their depth. The sampling of data within fjords is therefore carried out in a more *ad hoc* manner, with many smaller teams and experiments creating disparate datasets that suffer from issues of FAIR (Findability, Accessibility, Interoperability, and Reuse) data, and some fjords are sampled much more heavily than others
(Bischof et al., 2019a). This is a known issue and there has already been work done to create unified datasets for specific aspects of Arctic fjords (e.g. physical oceanography via UNIS CTD database; Skogseth et al., 2019; https://data.npolar.no/dataset/39d9f0f9-af12-420c-a879-10990df2e22d). There is not yet however a unified dataset that provides data for investigating the range of possible relationships throughout the entire socio-ecological fjord system. The dataset detailed in this report aims to address this shortcoming.


The combination of the many different socio-ecological datasets is not as simple as identifying the sources of *in situ* data and putting them together into a folder. With the exception of a network of meteorological stations operated by the Norwegian Meteorological Institute (frost.met.no), and a few well established instrument installations (e.g. Fischer et al., 2020), there are precious few long running time series collected within or adjacent to the fjords of the EU Arctic. Because maintaining
these stations in the Arctic is so expensive, and because the hostility of the terrain dramatically limits their potential size/scope, there is always much more demand for research support than can be given. This is managed in part by running seasonal projects (1-3 month duration). In order to extend the time series for these projects research teams may occasionally leave instrumentation in the field to continue sampling until the teams arrive the following year and they begin a new (though usually similar) project (e.g. Bartsch et al., 2022). While this has proven to be an effective strategy for optimising
the available fieldwork time in the Arctic, it effectively creates many short time series, with a range of interoperability issues. There are now many well funded international projects and research institutions that are working to close knowledge gaps in Arctic systems, but they tend to continue to produce these smaller datasets.

A primary consideration therefore during the creation of an Arctic fjord dataset designed to allow for the investigation of the
full socio-ecological system within fjords is how to combine many spatially and temporally disjointed datasets when they may not have the same units of measurement or have otherwise not been sampled with comparable methodologies. To begin to address this issue a panel of experts in a range of natural and social science fields identified the most relevant aspects of Arctic fjord socio-ecological systems (Schlegel et al., in review). The structure of these systems were organised by: *category*

-> *driver* -> *variable*. For example, the proportion of sea-ice cover within a fjord is a *variable* of the *driver* sea-ice, which is
in the *category* of the cryosphere. This structure was used to guide the collection of data and to organise how the many small
yet very important datasets in the Arctic were amalgamated. This structuring of the available data also allowed for better
management and conversion of the different units and methodologies into a project-wide standard.

In the following text we first explain why certain study sites were focussed on when collecting the datasets that contained the
variables of interest. We then document the methods by which these datasets were accessed, assembled, and quality
controlled to create a final product (Schlegel & Gattuso, 2022). A basic summary is then presented, based on the different
categories of the data (e.g. cryosphere, biology, etc.). To demonstrate possible uses of this dataset, drivers with known
important relationships (i.e. seawater temperature and sea-ice cover) are compared. Lastly, these relationships are projected
into the future with the aid of model data that are external to the dataset documented here.

## 2 Methods

### 2.1 Study sites

Many of the long running continuously sampled time series in EU Arctic fjords that contain the data of interest for socio-
ecological systems are located in one of seven study sites. Across these sites one also finds a gradient in the effects of the
changing climate on the Arctic cryosphere and all downstream processes. The future of what much of the Arctic may look
like is represented by fjords in mainland Northern Norway, in this case Porsangerfjorden (Fig. 1). This fjord completely
lacks a glacier, and frequently lasts the winter with little to no sea ice cover. There are then fjords further to the North on the
Svalbard archipelago (e.g. Kongsfjorden, Isfjorden, and Storfjorden), which do have glaciers and sea ice, but at variously
advanced rates of melt. Most projections show these fjords resembling those on the mainland at some point within the
century (Hop & Wiencke, 2019). Different again from the Svalbard fjords are those found in Greenland. With the east coast
(e.g. Young Sound) currently less influenced by warming coastal waters than the fjord systems on the west coast (e.g.
Qeqertarsuup Tunua and Nuup Kangerlua). Using the names of these seven sites (accounting for various different spellings)
as well as their geographical coordinates, the databases detailed below were queried to create individual data collections per
site. The main city/research station was also used in queries for data at Kongsfjorden (Ny-Alesund/Ny Alesund/Ny-
Ålesund), Isfjorden (Longyearbyen), Young Sound (Zackenberg), Qeqertarsuup Tunua (Qeqertarsuup), and Nuup Kangerlua
(Nuuk). The search parameters were not case sensitive.



**Figure 1:** *Decadal* trends in sea surface temperature (SST) throughout the Arctic waters surrounding the seven study sites (roughly 60° W - 60° E and 60° N - 90° N) and *annual* trends in sea ice cover. The colour of the pixels in the central panel show the decadal rate of change from a simple linear model of the annual average temperatures during the period 1982-2021 from the daily NOAA OISST 0.25° gridded product (Huang et al., 2021). The location of the study sites are denoted with coloured points and are shown with colour-coordinated inset windows. The rates of change in sea ice cover (days per year) for each study site were determined with a simple linear model on the number of open water days per year from the MASIE ~0.04° gridded product (NSIDC, 2022). The thin purple contours found in some windows show the 0 days/year contour line, while pixels outside of the study site are shown as black. Note that the size of the study sites differ and this is not accurately reflected by the size of the windows.

## 2.2 Categories -> drivers -> variables

Due to the diverse range of avenues of inquiry one must consider when amalgamating data across the scope of a socio-ecological system, it was necessary to establish a consistent terminology. Each individual variable of measurement of the




natural and social world (e.g. the presence of ice, tourist arrivals, or nitrate concentration) was characterised into one of 14 drivers (*sensu* Möller et al., 2022), with each of these grouped into one of five categories (Table 1).

**Table 1:** Categories and drivers into which all data points in this dataset are classified. The categories (top row) are: cryo = cryosphere, phys = physics, chem = chemistry, bio = biology, soc = social. The drivers are: sea ice = sea ice cover, glacier = glacier mass balance, runoff = terrestrial runoff, sea temp = seawater temperature, light = spectral radiation (PAR + UV-B), carb = carbonate system, nutrients = nutrients, prim prod = primary production, biomass = biomass, spp rich = species richness, gov = governance, tourism = tourism, fisheries = fisheries.

| cryo | phys | chem | bio | soc |
|------|------|------|-----|-----|
| sea ice | sea temp | carb | prim prod | gov |
| glacier | salinity | nutrients | biomass | tourism |
| runoff | light | | spp rich | fisheries |


The list of 14 drivers was not initially evident, nor was there a consensus on them from the start. At the outset of the project a long list of relevant variables was agreed upon and links to the necessary datasets were provided when possible. When no link was provided, a series of data sources (see Section 2.3) were queried using keywords or units of measure (e.g. sea water OR °C) from the list of variables. While all of the data originally identified were aggregated, a literature review performed

for this same project revealed that the important interactions within socio-ecological systems would be better expressed as broad drivers, rather than individual variables (Schlegel et al., in review). In reaction to this, further pruning of the dataset outlined here revealed that many of the variables from the initial list had little to no available data. After a couple rounds of editing and the final list of five categories, 14 drivers, and the variables therein were established. Thanks to the companion review paper (Schlegel et al., in review), it was also possible to determine which relationships between drivers are the most

important, and what the direction of those relationships are. It is these important dependent relationships that are used to demonstrate the utility of the product (see Section 4 and 5). Finally, it should be noted that not all variables have equally accessible amounts of data, and the collection of data was heavily skewed in favour of the well sampled variables of seawater temperature and salinity (Fig. 2). A total of 7,564,441 data points have been collected, with nearly half being seawater temperature (3,606,138), and the other half salinity (3,482,342). Of the 1,565 datasets that have been collected, 880

contain seawater temperature data. Of the 107 different variables, 81 of them come from a single dataset. These primarily being variables for biology, cryosphere, and social drivers (Fig. 2).





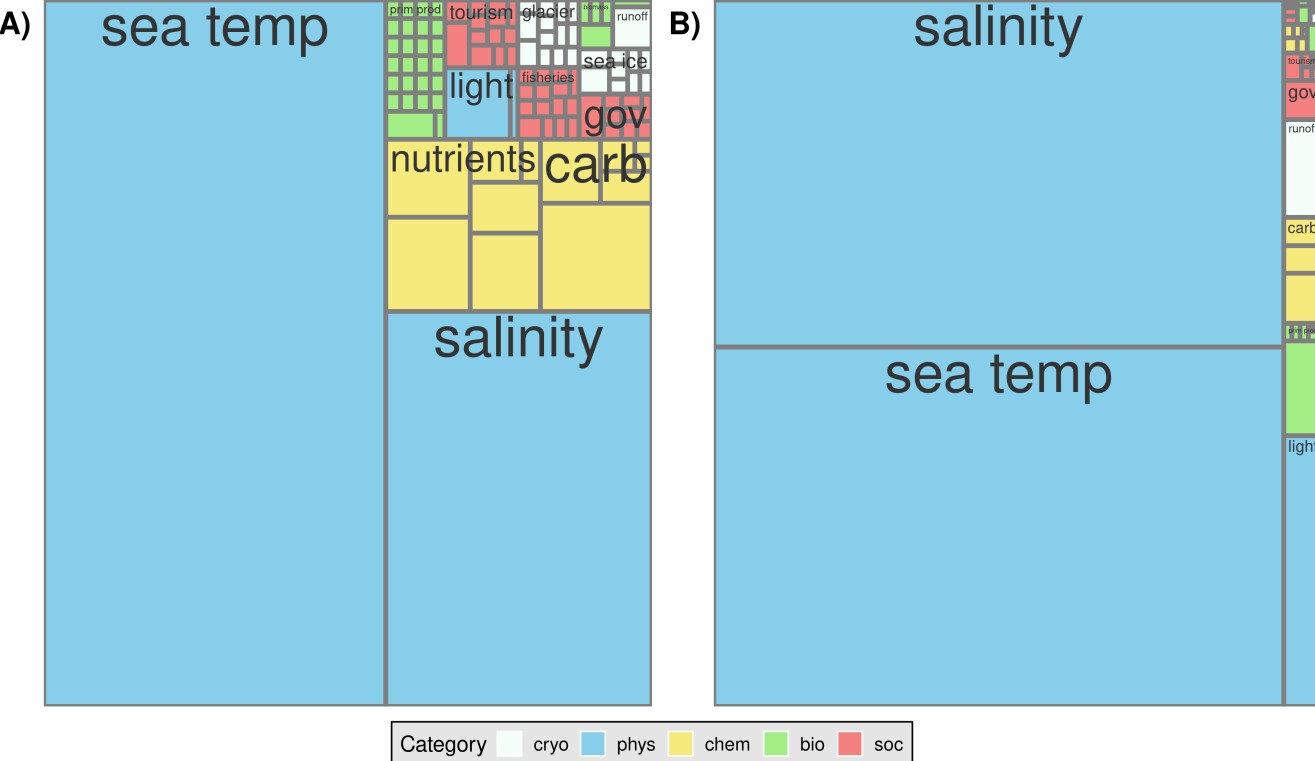

**Figure 2:** Square tree plot showing the relative presence of the data collected for this dataset. Each box represents one variable. The clusters of variables per driver are labelled, with the colour of the boxes indicating the category of the data. If the number of data points for the variables constituting a driver are not numerous enough, no label is plotted. Definitions for the contractions used here (e.g. "carb", "sea tep", etc.) are given in Table 1. Panel A) shows the relative count of datasets containing the indicated driver. Note that these boxes are not independent of one another because a single dataset could potentially contain multiple drivers. Panel B) shows the relative count of individual daily data points per variable. These boxes are independent of one another.

**2.3 Data sources**

The vast majority of the data aggregated for this dataset were publicly available and accessed via four data repositories: 1) PANGAEA, 2) The Norwegian Polar Data Centre (NPDC), 3) The Norwegian Marine Data Centre (NMDC), and 4) Greenland Ecosystem Monitoring (GEM) (Table 2). A description of these repositories and how the datasets within each were accessed are detailed in the following subsections. Other notable databases that provided important access to data are also mentioned below.

**Table 2:** The 'Total' count of datasets identified per site that contain data for the 14 drivers identified in this study. The count of datasets contributed from the four largest sources are listed in individual columns: PANGAEA, NPDC (Norwegian Polar Data Centre), NMDC (Norwegian Marine Data Centre), GEM (Greenland Ecosystem Monitoring), with the 'Other' minor sources combined into one column. The number of datasets containing data for a driver within one of the five categories are also listed: cryosphere (cryo), physics (phys), chemistry (chem), biology (bio), and social (soc) are also numerated. Note that a single dataset may contain data for multiple sites or categories.

| Site | Total | cryo | phys | chem | bio | soc | PANGAEA | NPDC | GEM | NMDC | Other |
| --- | --- | --- | --- | --- | --- | --- | --- | --- | --- | --- | --- |





| Kongsfjorden | 102 | 5 | 79 | 29 | 5 | 1 | 85 | 10 | 0 | 0 | 7 |
|---|---|---|---|---|---|---|---|---|---|---|---|
| Isfjorden | 110 | 4 | 97 | 17 | 3 | 2 | 98 | 4 | 0 | 2 | 6 |
| Storfjorden | 49 | 8 | 36 | 10 | 0 | 1 | 41 | 4 | 0 | 0 | 4 |
| Young Sound | 16 | 7 | 3 | 5 | 3 | 0 | 1 | 0 | 12 | 0 | 2 |
| Qeqertarsuup Tunua | 284 | 2 | 279 | 22 | 1 | 0 | 276 | 0 | 4 | 0 | 3 |
| Nuup Kangerlua | 458 | 7 | 443 | 36 | 4 | 0 | 445 | 0 | 9 | 0 | 4 |
| Porsangerfjorden | 243 | 2 | 239 | 3 | 1 | 0 | 196 | 0 | 0 | 42 | 4 |

### 2.3.1 PANGAEA

The PANGAEA data portal (https://pangaea.de) hosts a very large collection of datasets produced primarily via earth system
research. It is an open access portal with only some datasets under password protected embargo while the authors are waiting
for a corresponding research article to finish the publication process. The administrators of this portal provide an API
through which one may programmatically interrogate the entire database of a few hundred thousand datasets using boolean
search operators and keywords. While it must be noted that this data portal does not specialise in Arctic data, it is possible to
filter data within a lon/lat range, ensuring that the search results remain relevant. For this particular project the R package
'pangaeaR' (Chamberlain et al., 2021) was used. Through the initial search process, 14,063 datasets were identified as
potentially within the scope of the search for the key drivers and study sites. This first process was based primarily on which
datasets were geolocated within the bounding boxes covering the seven sites (Fig. 1), as well as filtering out datasets that
were specifically bathymetric, terrestrial, or aerial in nature. After downloading the datasets and amalgamating them, the list
of parameters for PANGAEA data were consulted and those applying to the 14 drivers determined for this dataset were used
as a second filter on the downloaded data. Through this process it was determined that 840 of the PANGAEA datasets would
be aggregated, at least in part, with the final dataset.

### 2.3.2 NPDC

The Norwegian Polar Data Centre (NPDC; https://data.npolar.no) is designed to cater to the needs of the Arctic research
community and specifically hosts datasets related to research conducted for, or funded by, Norwegian institutions. This
portal provides a more classic user interface in which one has a search bar that understands boolean logic. Because the
indexing of datasets on this website is tailored to Arctic research, it was not necessary to use the keywords for the drivers in
searches. Rather it was sufficient to search just for the names of the study sites. 11 datasets were downloaded from this
website and all of them were aggregated into the final dataset.



### 2.3.3 NMDC

The Norwegian Marine Data Centre (NMDC; https://www.nmdc.no/) is also designed to host Norwegian data, but focuses on the marine realm, and not necessarily Arctic. This is not however an issue as the database interface combines a keyword search bar, common categories that can be ex/included via radio buttons, as well as an interactive map that allows one to filter via spatial domain. In this way one can rapidly and accurately search for datasets containing drivers of interest within the predetermined spatial domains of the seven sites chosen for this project. 44 datasets were downloaded from this portal,

all of which were included in the final dataset.

### 2.3.4 GEM

The Greenland Ecosystem Monitoring Database (GEM; https://data.g-e-m.dk/) focusses on the management and dissemination of data relevant specifically to the three Greenland study sites identified for this project. This database is therefore oriented around inquiries into one of these three sites, and while a search bar is available, it is generally more direct

to follow the links provided for the individual sites and to use the file structures listed therein to find datasets of interest. Overall, 31 datasets were downloaded from this portal, however, due to the data portal requirements for acknowledging the use of each unique download, it is not possible to include these datasets in the final dataset presented in this paper.

### 2.3.5 Additional sources of note

The Svalbard Integrated Arctic Earth Observing System (SIOS; https://sios-svalbard.org/) is effectively a meta-search

database of other Arctic databases that contain datasets specifically of interest to researchers on Svalbard. This data portal provides an advanced user interface, similar to the NMDC, in which a range of criteria may be imputed in some way in order to limit the resulting output. After searching through the NPDC and NMDC databases, SIOS was used to perform a meta-search of many additional databases to catch anything that was not hosted on the two primary Norwegian sites. Several datasets were discovered through this method, all of which were included in the final dataset.


Another database with a strictly Arctic focus that provides publicly accessible data is Environmental Monitoring of Svalbard and Jan Mayen (MOSJ). This database has a drop-down tab menu that allows users to select broad categories like Climate -> Ocean. And therein one can select from several variables such as sea ice extent or sea level. One then directly downloads these data as.csv files. This website was of particular importance for direct and useful glacier mass balance data. But it also

provides a full range of variables from the cryosphere, physics, and biology categories. In total 2 datasets were downloaded here, all of which were included in the database.

Governance data were provided exclusively via the national statistics websites of Norway (https://www.ssb.no/en) and Greenland (https://bank.stat.gl/pxweb/en/Greenland/). The Norwegian statistics website focuses more on the national



concerns of an economy oriented around more developed service industries, and therefore has fewer resources available for dedicated inquiry into the human impact on the marine realm. The Greenland national statistics website, in contrast, focuses more on the importance of the marine realm to the economy and therefore has a deeper range of available statistics of interest for the effect of governance on other drivers in Arctic fjord socio-ecological systems. Overall 8 datasets were downloaded for Norway, 16 for Greenland, and all were included in the database. It must be noted that the spatial scale of

these data is much greater than the other categories. For example, the 'site' of collection for a statistics is usually an entire province, not a single fjord. Where possible, the national scale sites are associated with their local scale fjord (e.g. Nuup Kangerlua is within the Sermersooq Municipality).

There are also a few very large datasets of interest to this project who are themselves an amalgamation of existing smaller

datasets. The first of these is the UNIS database, which is a collection of all of the moorings found throughout western Svalbard and consists primarily of salinity and temperature measurements, which partially explains the dominance of these two variables in the dataset (Fig. 2B). The other two datasets, SOCAT (Surface Ocean $CO_2$ Atlas; Bakker et al., 2016) and GLODAP (The Global Ocean Data Analysis Project; Lauvset et al., 2022), focus more on the carbonate system of the ocean. These are global products, but only the Arctic region encompassing the seven study sites (~60°W to 60°E, ~60°N to 90°N)

has been amalgamated into the dataset for this project. Finally, the Norwegian Meteorological Institute (https://www.met.no/en) provided 13 very long and high quality multi-variable atmospheric time series generated by long-running MET stations. Unfortunately the final dataset for this project was limited to focussing directly on the marine realm, not accounting for the atmosphere, so these time series were not amalgamated.

**2.4 Data assembly**

Once the data portals had been thoroughly interrogated and the files had been downloaded and saved according to their data sharing permissions, they were combined into a single data product. Each of the data portals outlined above have their own requirements for the data they host, with some portals being more strict than others. Much of the aggregation of the hundreds of different datasets was aided by the very strict quality control for data hosted on PANGAEA. All of these datasets were first aggregated during the download process into a PANGAEA specific format, which was close to the final project-wide

standard. The other data portals allow for a wider variety in which the raw data within the dataset may be oriented, as well as the different file types within which those data are stored. Using the R language (R Core Team, 2022), a series of scripts were written to create a pipeline that loaded first all of the PANGAEA files, then the individual files from the other data portals, before combining them into a single shared project-wide standard based on tidy data principles (Wickham, 2014). Each datum in the dataset therefore has the same corresponding columns of meta-data: (1) download date, (2) URL, (3)

citation, (4) type of data (e.g. *in situ* or remotely sensed), (5) site (mostly one of the seven study sites; Fig. 1), (6) category, (7) driver, (8) variable (9) longitude, (10) latitude, (11) date of sampling, and (12) depth of sampling. Where possible, the URL provided for the data is the link to its digital object identifier (DOI) page. Importantly, all values in the dataset are



numeric so that they can be listed in one single column that extends along millions of rows of data. It should therefore be noted that non-numeric data were not amalgamated, and that data spanning multiple dates or depths were taken at either the mean date/depth or the max date/depth as would be appropriate for the data type. Once a single row of meta-data was finalised for each datum, any duplicate rows (i.e. hourly data) were averaged to a single value. This important decision was made because it was determined that for a data product of this scale it would not be beneficial to have sub-daily data. Indeed, seawater temperature already dominates this data product, and if it was left at its native resolution (often hourly values for moorings), then this would effectively be a seawater temperature/salinity product with <0.1% of the space dedicated to the other 12 drivers.

## 2.5 Quality control

Because all of the data aggregated for this dataset were taken from published sources, it was determined that they should not require the application of rigorous quality control (QC). Therefore the primary function of QC for this process was to ensure that the data aggregated into the final product could be classified into one of the 14 drivers identified as being particularly relevant to Arctic fjord socio-ecological systems. This was necessary in part because many of the datasets listed at the outset of the project contained data for drivers outside of the final 14. So any data falling outside of this narrowed scope were filtered during the amalgamation process. This was relatively rare.

Even though this dataset is composed almost entirely of published data, it was noted while performing the example analyses below (Sections 4 and 5) that some issues persisted. When the issue was simply an anomalous data point (e.g. a negative chlorophyll value) it was removed and the data analyses carried on. However, there were cases when systemic issues were identified in a dataset (e.g. consistently low salinity values). When possible the contact person for the dataset was notified and corrections were made according to their expert advice

## 3 Data summary

One would generally assume that the availability of data within the Arctic would be highly seasonal, but with the exception of the cryosphere, this is not the case (Fig. 3). The coverage of sea ice data is much lower in Spring and Summer because a complete lack of sea ice cover is generally calculated as a missing value, rather than a 0. Curiously, glacier mass balance data are missing in the Winter. Upon closer inspection it was discovered that this is because sampling tends to end in September and resume in April. We also note that the values for social drivers (i.e governance, tourism and fisheries) are so low because these data are only available at monthly or annual rates, whereas the data from the other categories (i.e. not social data) are available at daily rates.

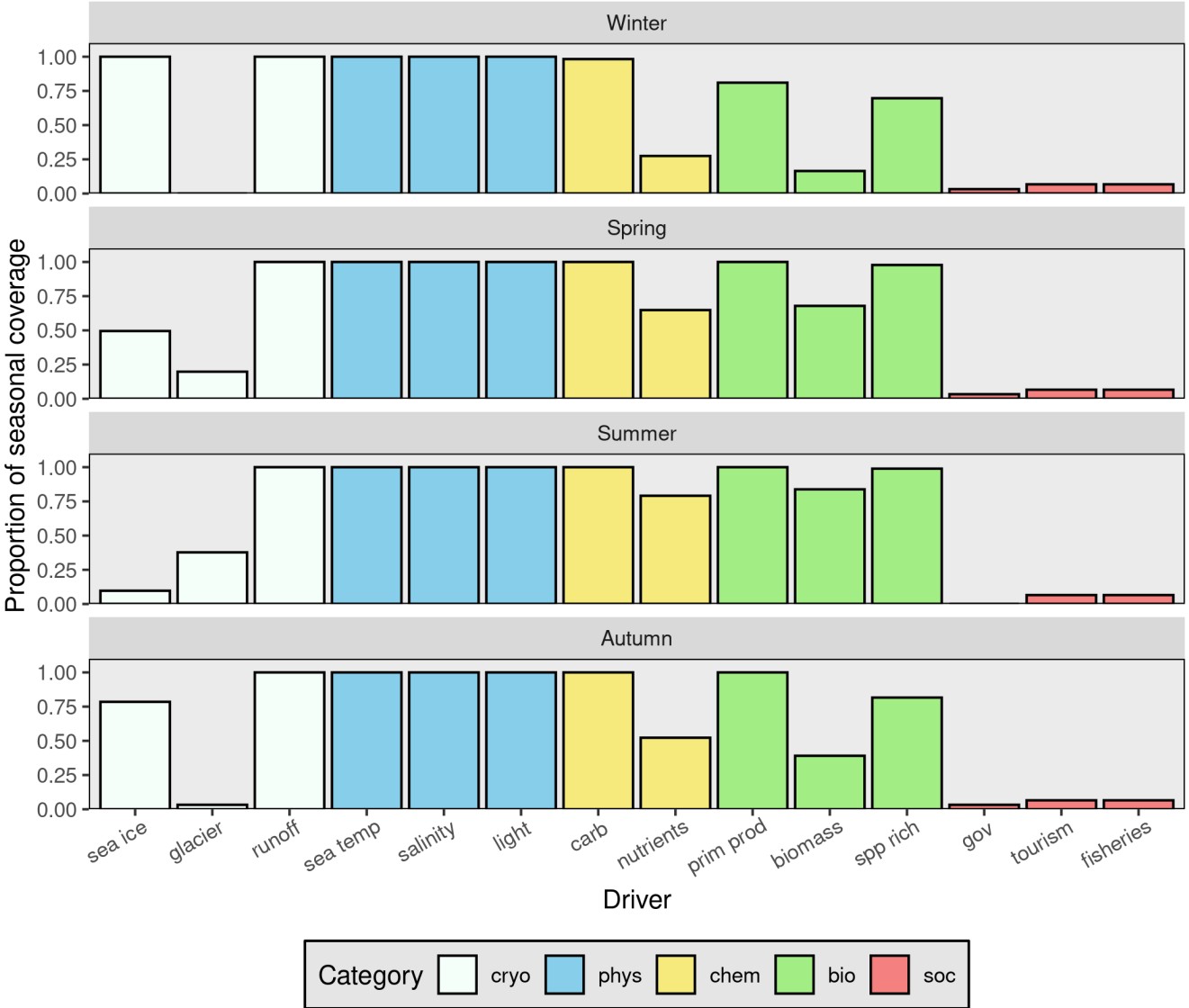

**Figure 3:** Summary of available data for the drivers identified for this project. Each panel shows the coverage per driver for the given
season, with Winter containing January, February, and March. Seasonal coverage is calculated by the total number of unique days-of-year
within a season that have at least one datum present for the given driver. The colours of the bars indicate the category to which the drivers
belong. The very low social driver values indicate that these data are only available at a monthly or annual resolution, not daily like the
other drivers.

Besides differences in seasonal coverage, some drivers have data available for a much longer period of time than others (Fig.
4). Seawater temperature and salinity are once again the most well sampled of the drivers, with data starting in 1876.
Somewhat surprisingly though, nutrient data have been sampled since 1934. After that began the consistent measurement of
carbonate chemistry, terrestrial runoff and glacier mass balance data in 1957, 1957, and 1967 respectively (Fig. 4). The



measurement of the rest of the drivers tends to begin in the 1990's to 2000's. We may also see that data for all of the drivers

275    tend not to be present at all seven study sites at the same time (Fig. 4). The data available for the 14 drivers within each of

the five categories are summarised in the following sub-sections and any specific filtering or unit conversions made are

detailed.









**Figure 4:** The annual count of data for each driver across the seven study sites in this data product. The colour of the bars shows how many total sites have data present in a given year, with the height showing the total count of the data. Note that the y-axes differ for each driver, with seawater temperature and salinity being much greater than the others. The year with the highest count of data for each driver is labelled. There is a break on the x-axis at 1957 denoted by a vertical dotted line. Seawater temperature and salinity have data going back from 1956 to 1876, and the sum of all of these annual values are shown as bars to the left of the dotted line.

## 3.1 Cryosphere drivers

Cryosphere data are readily available throughout the Arctic, though usually not at a daily resolution. Measurements of winter ice cover are generally available for all but the western Greenland sites. The glaciers for Svalbard and eastern Greenland have annual measurements available in August/September for Svalbard, and April/May for eastern Greenland. Conspicuously, there are no measurements of sea ice across sites using the same methods/units. Comparisons of *in situ* sea ice cover across the Arctic are thus not currently possible. The Global Runoff Data Centre (GRDC) provides a global database of river discharge values, though their data sharing restrictions prevent us from aggregating the data here. Likewise, there are many discharge values available for Greenland via GEM. The terrestrial runoff data that were subset from the larger GRDC database were determined by spatially filtering the time series whose lon/lat values fell within the bounding boxes of the study sites (Fig. 1). The number of datasets providing cryosphere data are relatively low compared to those for chemistry or physics data, but higher than for biology and social drivers (Table 2).

Due to its broad importance for the understanding of change within Arctic fjord socio-ecological systems, sea-ice cover is one of the two drivers in this data product for which remotely sensed data were included. Pixels were extracted from the MASIE 4 km resolution ice cover product (NSIDC, 2022) for the seven study sites (Fig. 1), and average daily sea-ice cover values were created from 2006-2021. This created only one additional time series per site, thereby avoiding to overrepresent remotely sensed data in this data product, which aims to be a collection of primarily *in situ* data.

## 3.2 Physics drivers

Data measuring the physical properties of the EU Arctic are the most readily available (Fig. 2, Table 2). Seawater temperature measurements are available at daily to monthly resolution for all sites, with the most frequent taken at Kongsfjorden, and the least at Young Sound, where there are no values during the winter months. Salinity data are often sampled alongside seawater temperature, and so their availability largely matches the former. Light data, while important, are much more difficult to come by. There are many daily values available at Kongsfjorden, but only for one or two years. There have been a few years of summer light measurement in Young Sound, and the western Greenland sites have enough data to create a rough monthly climatology. No data have yet been sourced for northern Norway.

The importance of seawater temperature within Arctic fjords made it the second of the two drivers for which remotely sensed data were sourced. Data for the entire bounding box of the study area (Fig. 1) was subset from the daily NOAA OISST v2.1 0.25° resolution product (Huang et al., 2021). The pixels within the bounding boxes for the seven study sites





were combined into a single daily time series from 1982 to 2021. However, while a resolution of 0.25° may be sufficient for ocean scale studies, for many of the fjords in this data product, this resolution is too coarse. Therefore, seawater temperatures for each site were also sourced from v2.1 of the Climate Change Initiative (CCI) daily 5 km resolution product produced by the European Space Agency (Merchant et al., 2014). Time series were created for the period 1982-2020 by averaging the daily values over all pixels found within the bounding boxes for the sites (e.g. Fig. 1).

### 3.3 Chemistry drivers

The chemical composition of seawater is generally well sampled in the Arctic with the study sites having data available for the carbonate system for all months of the year (Fig. 3). Daily carbonate system data are available for several years in Kongsfjorden, and to a lesser extent in Nuup Kangerlua. Nutrient data availability is greatest during the ice-free months (Fig. 3), particularly in Kongsfjorden and Young Sound. While less frequent, the western Greenland sites also have data available for much of the year. Datasets providing chemistry data are the second most numerous after physical data (Table 2).


The filtering and grouping of variables for the chemistry drivers required more consideration than the previously described categories because each of these drivers contained variables that were notably different from one another. For example, while sea ice cover data might be in units of % or km2, one can still filter through datasets for any reference or variable name containing 'ice'. However, the carbonate system encompasses the partial pressure of CO2 ($p$CO2) in seawater, total

alkalinity (TA or $A$T), dissolved inorganic carbon (DIC or $C$T), pH, and the saturation state of calcium carbonate. Likewise, nutrients contain nitrate (NO3), nitrite (NO2), ammonium (NH4), phosphate (PO4), and silicate (SiO4). All of the variables for these two drivers tend to come in a variety of units of measurement, so it was necessary to choose a standard unit and convert data as necessary. For the carbonate system this was either in units of µmol kg-1 (TA, DIC), µatm ($p$CO2), and total scale (pHT) when possible (unfortunately many pH values have an unknown scale, which is noted in the units for the

variable). For the nutrients, all values were converted to µmol l-1.

### 3.4 Biology drivers

Datasets providing data for biology drivers are not numerous (Table 2). Indeed, no datasets were identified for Storfjorden or Porsangerfjorden, with only primary production data in August available at Qeqertarsuup Tunua. Nuup Kangerlua is the only site with data available for all of the biology drivers for all months of the year, with Young Sound having all drivers during

some ice-free months. Isfjorden has primary production data available over the full calendar year with some daily datasets available. Note that seabird data exist for much of Svalbard (e.g. https://data.npolar.no/dataset/0ea572cd-1e4c-47a3-b2a5-5d7cc75aaeb4), but these were considered to be outside of the scope of the marine data collected for this dataset.

It is important to note that while the rate of primary production is known to be a very important driver in Arctic fjords, data

for the direct measurement of this driver are almost non-existent, with the exception of Young Sound, thanks to a recently





published study (e.g. Holding et al., 2021). Otherwise most primary production values come from personal communications (e.g. Hop et al., 2002) or from a couple of historic data points (e.g. Eilertsen et al., 1989). To address this shortcoming, the data collected for chlorophyll (Chl a [µg l-1]) were grouped with primary production. This is a potentially controversial choice, but was made because it was necessary to make additional compromises for the other biology drivers, which left

primary production as the best classification for Chl-a data. One could argue that these data would be better placed in the biomass driver however, data for this driver were also lacking in public availability. Because of this, the data classified here as biomass are species survey data when the units are reported in individuals m-3 or cells m-3. While not ideal, providing these data to the community still allows for researchers who know which species they are looking for to readily access them via this data product and to perform the biomass calculations for themselves. It is beyond the scope of the data amalgamation

for this product to perform these calculations for the 751 species that have data available in this product. Lastly, that brings us to species richness. No publicly available data exist that report on this driver directly. To address this we removed the units (e.g. individuals m-3) from every measure of a species and tabulated them per site, day, and depth to get the count of species, which then forms the basis of what could be an investigation into species richness. The presence of the individual species per site, day, and depth were also maintained so that researchers can access this information. Again it was

determined to be outside of the scope of the data amalgamation for this project to perform analysis on these data, such as calculation of Shannon Wiener diversity indices.

### 3.5 Social drivers

Of all the categories of drivers identified in this study, datasets for social drivers were the most difficult to source. This is primarily due to the fact that there are no applicable social datasets on PANGAEA, which is by far the largest provider of

data in this project (Table 2). Of the hundreds of datasets sourced, only 28 of them provide social data. Three provide monthly tourism values over the past several years and for Kongsfjorden and/or Isfjorden (Table 2, Fig. 3), one for ship AIS data in western Svalbard, 10 for monthly governance statistics for Greenland, and another 8 for Norway. Fisheries are a very well quantified driver, with a very well established body of statistical analyses for comparison with the natural world. Six such datasets were sourced via national statistics websites for Greenland and three for Norway. There are 19 fisheries

datasets available via the IMR site (https://gbif.imr.no/ipt/), three of which have been amalgamated.

More so than with the other categories of data, there is overlap in the variables for the drivers within this category. For example, boat traffic within the fjords is an important social consideration, but must be classified either into the tourism or fisheries drivers depending on the ship in question. The variable names are otherwise the same, which required that the

reference in question be consulted in order to accurately rename them (i.e. 'vessels - tourism [n]' vs 'vessels - commercial [n]'). The division and regrouping of these variables was by far the most time consuming of all of the categories due to how many small exceptions there were.



## 4 Relationships between drivers

In order to illustrate the potential uses of this dataset, a comparison of the different drivers is outlined below. This is not an
exhaustive comparison, nor is it meant to be proscriptive on the use of these data. That would depend on the question(s)
being asked by a given researcher. It must also be noted that the values presented below do not necessarily reflect the
changes that exist within the fjord as large aggregations of the data have been made in the interest of simplifying the analysis
in order to perform it across all of the categories/drivers of data. It would also not be productive or useful to compare every
driver in this dataset to each other. Rather it is necessary to follow a guiding principle for which drivers are compared and
why. This is found in Fig. 2 of Schlegel et al. (in review), which shows the key relationships between drivers and the
direction of their interactions, as determined from the literature. Because the aim of this dataset is to allow for investigations
of the interactions between drivers within a given Arctic fjord socio-ecological system, we did not compare data across sites,
but we did compare the interactions found within each site to those in other sites. In order to maximise the amount of
comparisons that can be made for drivers within fjords, the data have been binned into a few consistent depth ranges, and
averaged into monthly means. Time series with only annual values were not used as these created skewed comparisons
against other drivers for just the months of January or December accordingly. When data were available for two different
comparisons at multiple different depth ranges (e.g. surface sea ice, and seawater temperature at 50 m) all of the possible
depth range comparisons were made. Generally speaking, comparisons of data at the same depth will be the most interesting,
but there are exceptions to this that are noted below. Comparisons were only made when there were at least three months of
overlapping data available, and only data from 1982 to 2020 were used as this is the period available for the CCI SST
product.

It must be noted that even though the methodology used for data comparison is coarse, there are still many drivers with
either no overlapping monthly values, or only a couple of sites that have overlap. For many sites there are almost no drivers
that overlap with anything other than seawater temperature (Table 3). Of the 217 relationships that exist between the
variables within drivers, only one was able to be quantified across all of the seven sites contained within this dataset (Table
3). 18 comparisons could be made between just two different sites, seven between three sites, and seven more comparisons
could be made between four or more sites.

**Table 3:** The drivers and variables with overlapping monthly data that could be compared within multiple sites. The independent
drivers/variables are shown in the columns with an 'x', and the dependent drivers/variables in the columns with a 'y'. The site count
column shows the number of sites within which the indicated comparison could be made. Note that for seawater temperature and sea ice
this includes remotely sensed data.

| driver x | driver y | variable x | variable y | site count |
|---|---|---|---|---|
| sea temp | sea ice | temp [°C] | sea ice cover [proportion] | 7 |

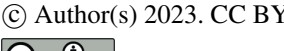



| runoff | salinity | Q [m3/s] | sal | 6 |
|---|---|---|---|---|
| runoff | sea temp | Q [m3/s] | temp [°C] | 6 |
| salinity | spp rich | sal | spp count [n] | 5 |
| sea temp | spp rich | temp [°C] | spp count [n] | 5 |
| runoff | light | Q [m3/s] | PAR [µmol m-2 s-1] | 4 |
| sea ice | light | sea ice cover [proportion] | PAR [µmol m-2 s-1] | 4 |

The relationship between seawater temperature and sea ice cover was somewhat consistent across all sites (except Young Sound), with an increase of 1° C equating to a mean change (± standard deviation) in annual sea ice cover of -12% ± 0.22% (Fig. 5). This relationship allows us to project this value into the future given different emissions pathways (Section 5). Another robust comparison available within this dataset is seawater temperature and species count, which shows a positive trend for most sites, with Young Sound again providing outliers (Fig. 5).






**Figure 5:** Boxplots showing the values for the slope of linear models that compare an independent and dependent variable within multiple sites, as shown in Table 3. The colour of the dots shows the site, which may have more than one dot as multiple depths were compared between variables.



## 5 Future change of drivers/relationships

The projections of the relationships found between drivers into the future were made possible with the use of the
NORWegian ECOlogical Model system (NORWECOM; Aksnes et al., 1995; Skogen et al., 1995; Skogen and Søiland,
1998). This model couples physical, chemical, and biological systems in the Arctic (as well as other regions) and contains
multiple representative carbon pathway (RCP) projections at multiple depths for five of the seven study sites at ~10 km
resolution, with projections beginning to change from historic data from 2015 onwards. The two missing sites are from West

Greenland, and while there are data available for Young Sound, none of it overlaps with the data present in this product.
Because this model is on a ~10 km grid it generally does not contain data within the fjords, excepting the larger Isfjorden and
Storfjorden. Therefore, one must note that the future projections of the driver relationships detailed below are generally for
data over the shelf mouth of the fjords, and not for the inner fjord processes.

The Arctic model contains six overlapping variables with the data product detailed here: seawater temperature, salinity,
$pCO_2$, nitrate ($NO_3$), phosphate ($PO_4$), and silicate ($SiO_4$). As a first step to see how similar the data between the model and
the amalgamated dataset are, the RMSE for monthly data were made between sites at the same depths (Fig. 6). We then
looked at the differences in the trends of the data where possible (Table 4).

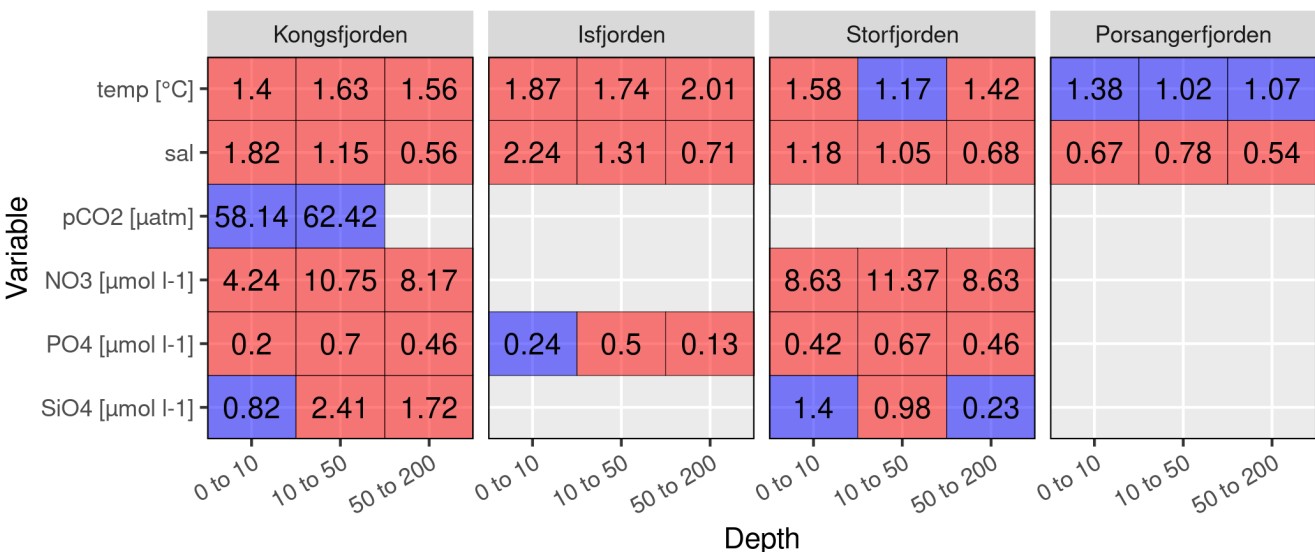

**Figure 6:** Heatmap showing the difference in mean monthly values between the NORWECOM model and amalgamated data at matching
depths. These values were calculated with the RMSE (root mean square of errors) statistic. Red squares show when the model values are
greater than the amalgamated data. Note that only RMSE for temperature comparisons to *in situ* sampled data are shown, not the remotely
sensed products.



**Table 4:** The projected changes to seawater temperature (temp [°C]) and salinity (sal) in four sites and 3 depth ranges with overlapping data to the NORWECOM model. Values shown are for decadal trends calculated over 1982-2020 for the *in situ*, OISST, and CCI seawater temperatures. The RCP projection trends are calculated from 2000-2099. NB: the decadal trends for *in situ* sampled temperature and salinity at most sites are clearly incorrect. This issue is caused by unequal sampling in the base data, which then expresses itself as anomalous values when grouped analyses are performed across different types of data (i.e. *in situ* vs remote vs model. One should therefore not take these values as representative of *in situ* changes in the fjords, but rather are indicative of challenges for using these data.

| site | variable | depth | in situ | OISST | CCI | RCP 2.6 | RCP 4.5 | RCP 8.5 |
|---|---|---|---|---|---|---|---|---|
| Kongsfjorden | temp [°C] | 0 to 10 | -0.7 | 0.15 | 0.39 | -0.02 | 0.06 | 0.18 |
| Kongsfjorden | temp [°C] | 10 to 50 | -0.48 | NA | NA | -0.02 | 0.08 | 0.21 |
| Kongsfjorden | temp [°C] | 50 to 200 | -0.27 | NA | NA | -0.02 | 0.08 | 0.21 |
| Kongsfjorden | sal | 0 to 10 | 0.32 | NA | NA | 0.01 | 0.02 | 0.02 |
| Kongsfjorden | sal | 10 to 50 | 0.02 | NA | NA | 0.01 | 0.01 | 0.02 |
| Kongsfjorden | sal | 50 to 200 | -0.01 | NA | NA | 0.01 | 0.01 | 0.02 |
| Isfjorden | temp [°C] | 0 to 10 | 0.22 | 0.22 | 0.42 | -0.01 | 0.07 | 0.2 |
| Isfjorden | temp [°C] | 10 to 50 | 0.08 | NA | NA | -0.01 | 0.08 | 0.22 |
| Isfjorden | temp [°C] | 50 to 200 | 0.16 | NA | NA | -0.01 | 0.07 | 0.21 |
| Isfjorden | sal | 0 to 10 | 0.77 | NA | NA | 0.01 | 0.02 | 0.02 |
| Isfjorden | sal | 10 to 50 | 0.48 | NA | NA | 0.01 | 0.01 | 0.02 |
| Isfjorden | sal | 50 to 200 | 0.18 | NA | NA | 0.01 | 0.01 | 0.02 |
| Storfjorden | temp [°C] | 0 to 10 | 3.49 | 0.1 | 0.29 | 0.01 | 0.08 | 0.2 |
| Storfjorden | temp [°C] | 10 to 50 | 5.96 | NA | NA | 0 | 0.07 | 0.18 |
| Storfjorden | temp [°C] | 50 to 200 | 3.04 | NA | NA | 0 | 0.06 | 0.15 |
| Storfjorden | sal | 0 to 10 | NA | NA | NA | 0.01 | 0.03 | 0.04 |
| Storfjorden | sal | 10 to 50 | NA | NA | NA | 0.01 | 0.02 | 0.02 |
| Storfjorden | sal | 50 to 200 | NA | NA | NA | 0.01 | 0.01 | 0.02 |
| Porsangerfjorden | temp [°C] | 0 to 10 | 0.35 | 0.32 | 0.1 | 0.03 | 0.1 | 0.24 |
| Porsangerfjorden | temp [°C] | 10 to 50 | 0.29 | NA | NA | -0.01 | 0.08 | 0.2 |
| Porsangerfjorden | temp [°C] | 50 to 200 | 0.63 | NA | NA | -0.01 | 0.08 | 0.19 |
| Porsangerfjorden | sal | 0 to 10 | -0.07 | NA | NA | -0.01 | 0 | 0 |
| Porsangerfjorden | sal | 10 to 50 | -0.01 | NA | NA | 0 | 0 | 0.01 |
| Porsangerfjorden | sal | 50 to 200 | 0.03 | NA | NA | 0 | 0 | 0.01 |

Even though the model data may be warmer than the *in situ* data for most sites (Fig. 6), the positive decadal trends in the remotely sensed data tend to be steeper than the RCP 8.5 projections. This is because the model data does not capture the cold winter temperatures as well as the remotely sensed data do. Therefore, even though the summer high temperatures in the model data may be increasing apace with the remotely sensed temperature, the model does not capture the same winter





time lows, thereby allowing the remotely sensed data to tilt upwards more aggressively due to the more pronounced winter time warming. One may note that there is a pronounced difference between seawater temperature and salinity trends between the model and amalgamated data. This is an artefact created by the coarse aggregation of these values in the amalgamated dataset across time (monthly averages), space (averages for all data points within a fjord), and depth (Table 5). These *in situ* values should therefore not be taken as indicative of any changes in the fjord. They are included here as a note of caution when aggregating these data further for analyses.

Even though we have demonstrated that the projections of the amalgamated data differ, sometimes dramatically, from the modelled data, the differences within the model for the three RCPs (i.e. 2.6, 4.5, and 8.5) are still interesting. In particular it is worth noting that the model projections for RCP 2.6 show that no further sea ice may be lost, and seawater temperature may no longer continue to rise (Fig. 7). In accordance with this, many of the current drivers of change within fjords would otherwise stabilise as these two drivers tend to be at the top of any downard cascade of changes. Were it only the case that human society was able to meet this scenario. Projections suggest that sea ice cover decreases dramatically with the somewhat extreme RCP8.5 scenario (Fig. 7). This is of course not surprising, and is very much consistent with the literature (Möller et al., 2022).

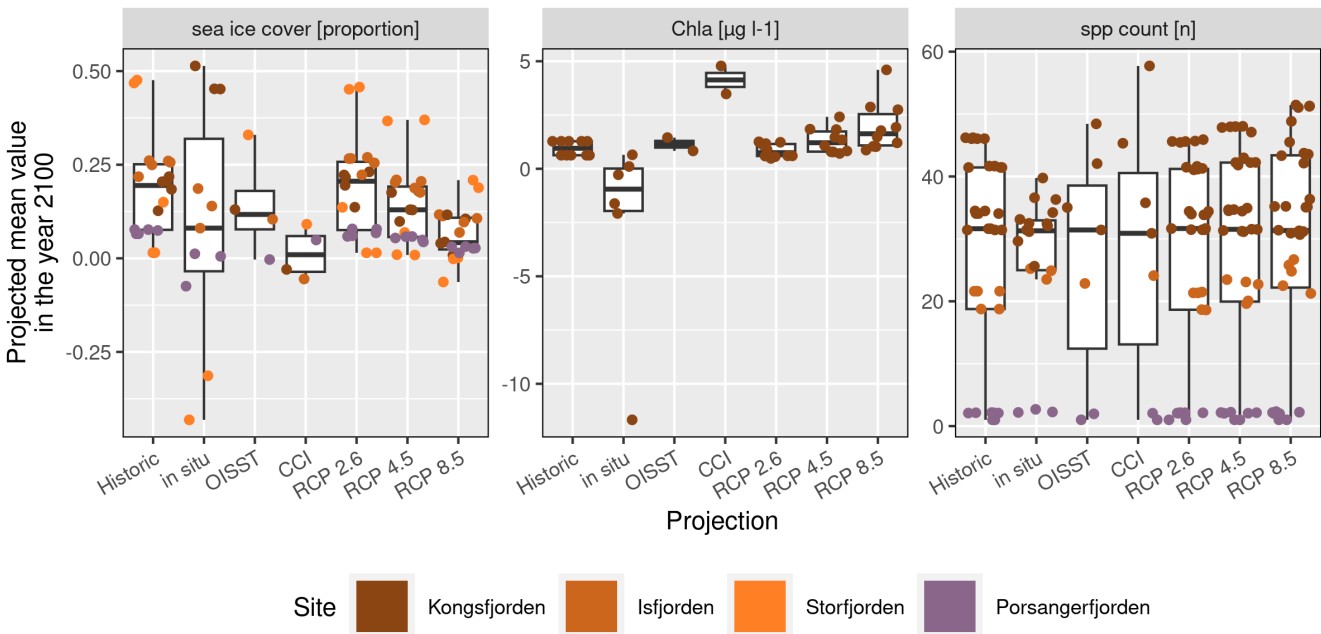

**Figure 7:** The historic values for sea ice cover, chlorophyll *a*, and species count in the amalgamated dataset across multiple sites and depths are shown in the first boxplot to the left of each panel. These values are then projected to 2100 using their relationships with seawater temperature determined from historic *in situ*, NOAA OISST, or CCI remotely sensed data. This was also calculated for the RCP 2.6, 4.5, and 8.5 projections from the NORWECOM model.



## 6 Code and data availability

As detailed above, certain decisions were made about which variables to group into which categories and drivers that may not be agreed on by all researchers. Regardless, due to the meta-data columns attached to each datum in this data product, it is possible to quickly isolate which data are of interest and extract them. For research projects making heavy use of data from
a limited number of references for data within this product, it is advised that these sources be cited in addition to the citation of this larger data amalgamation. This advice is similar to that for the use of data within the SOCAT (Bakker et al., 2016) and GLODAP (Lauvset et al., 2022) datasets.

The code written for the sourcing, cleaning and amalgamating of this dataset may be found on GitHub at: https://github.com/
FACE-IT-project/WP1. The code used for the figures and tables seen in this publication may be found at: https://github.com/ FACE-IT-project/WP1/blob/main/code/data_paper.R.

A meta-database providing a high level summary of the individually sourced datasets in this data product (i.e. not scraped from PANGAEA) is available here: https://face-it-project.github.io/WP1/metadatabase.html. A user interface to both the
final data product, and the raw version, may be accessed here: 193.50.85.104:4949/dataAccess/. The full data product (Schlegel & Gattuso, 2022) is hosted on Zenodo: https://doi.org/10.5281/zenodo.7472376  while the longer processing of minting a DOI via PANGAEA is being pursued.

There are two important data sources referred to in this manuscript whose data cannot be shared directly in this product due
to data access restrictions. Almost the entirety of terrestrial runoff data are found in the GRDC (Global Runoff Data Centre; https://www.bafg.de/GRDC/EN/04_spcldtbss/41_ARDB/ardb_node.html). Additionally, almost every source of coastal (non-social) data for Greenland is stored on the Greenland Ecosystem Monitoring database (GEM; https://data.g-e-m.dk/). The interested user must therefore go directly to these sites to source the relevant data.

## 7 Conclusion

The data product described in this report was assembled in order to address the needs of researchers who are investigating the interactions between, and changes to, key drivers within Arctic fjord socio-ecological systems. This was accomplished by sourcing and amalgamating numeric data available for 14 different drivers categorised into either the cryosphere, physical oceanography, chemical oceanography, biology, or social science. These data begin to have regular sampling as far back as the 1950's (or even 1900's), but more consistently from the 1990's forward. The distribution of the available data is not
equal between categories or drivers, with the majority of available data coming from seawater temperature and salinity.

There are enough overlapping data, both within and across the seven study sites, to allow for a range of transdisciplinary analyses. It must be noted however that most of these analyses across sites are aided by the inclusion of remotely sensed seawater temperature and sea-ice cover data. Without these the 'out-of-the-box' applicability of this amalgamated data
product to Arctic research would be reduced. Within the individual sites however there is enough *in situ* collected data for many interesting analyses.

The *in situ* collected data for many of the drivers in this data product required additional filtering (e.g. terrestrial runoff) and in some cases the conversion of the units of measurement (e.g. carbonate system and nutrients). Most of the data classified
into the biology drivers also required careful consideration as to how best to present the raw data to the user, while still maintaining a consistent project-wide standard for this dataset. This necessarily required the calculation of a species richness value, which was not present in any of the sourced datasets. Also calculated to improve the usability of this dataset were per-site time series of seawater temperature and sea-ice cover from remotely sensed sources.

This data product represents the first version of a data collection effort that will be ongoing for the next two years. Central to this process is the expansion of efforts to collect biological and social datasets, which are currently underrepresented here. The quantification of interview data is also something that is being investigated and will be addressed in the future version of this dataset. The monitoring of the primary online databases that have contributed to this project is ongoing and as datasets therein are updated, they will be amalgamated here. The future versions of this dataset will also be published on PANGAEA,
with backward references to this first version as is standard practice.

**Author contributions**

The paper was conceived by JPG, with the data collection and analyses implemented by RWS. Both authors contributed to the writing and editing of the manuscript, which was submitted by RWS.

**Competing interests**

The authors declare that they have no conflict of interest.

**Acknowledgements**

This study is a contribution to the project FACE-IT (The Future of Arctic Coastal Ecosystems – Identifying Transitions in Fjord Systems and Adjacent Coastal Areas). We gratefully acknowledge the input of FACE-IT colleagues for pointing to relevant data repositories. Special thanks are due to Morten Skogen for supplying the NORWECOM model data.



**Financial support**

FACE-IT has received funding from the European Union's Horizon 2020 research and innovation programme under grant agreement no. 869154.

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
