# Peer review of "A Dataset for Investigating Socio-ecological Changes in Arctic Fjords"

_Earth System Science Data, 2022_

## Referee Comment (RC2)

**Review of the manuscript: "A dataset for investigating socio-ecological changed in Arctic Fjords."**

This work manuscript presents the compilation of various data from Arctic Fjords, to investigate the socio-ecological changes in 7 Arctic Fjords. The manuscript is very clear and well written, the dataset is well presented and useful for several type of studies. However, this dataset is just a compilation of datasets already available in several databases, although I think that much more data are available but maybe not publicly. I suggest publishing it after major revisions. Please find my comments beneath.

**Main comment:**

My main comment is about the added value of this dataset. I understand that this is a compilation of already publicly available datasets. Although of interest for some studies I think such a compilation of data should aim for more than 7 fjords in the arctic, and rather try to compile all available data for Greenland and Svalbard fjords for example. A lot of data is available in a lot of fjords in Greenland, but maybe not easily available indeed. However, considering those data will add a lot of value to this dataset, that could be used to assess for example the impact of climate change on the Arctic Fjords.

**Minor comments:**

How do the authors define the Arctic? Is it the Arctic circle? I am a bit surprised that a fjord in Northern Norway is included in the manuscript. Please clarify.

L30: 'range of habitats for many important species': do you have any reference for that?

L34: Please delete 'extreme'

L37: This sentence argues that most of the sampling in the Arctic is performed by large cruise ships. I think it should be clarified here that you are talking about the coastal Arctic and I guess about surface data. In my knowledge, the central Arctic Ocean is mainly occupied by research ship so far, and most of the data come from autonomous platforms.

l.134: The manuscript explained the source of the data and the type of data, but for the ocean temperature for example, how are the data organized? Are they daily averaged or is it just a compilation of all available data in the fjord? For example, in Kongfjorden, several temperature datasets are available, moorings but also CTD from ships for example... Is that indicated in the dataset?

L.183: Did the author look at the arctic data center for more datasets? Is it on purpose that the fjords are only located on the Eurasian side of the Arctic. I could expect that quite a few data are also available in the Canadian Archipelago.

l.210: 'The first of these is the UNIS database': please add a reference

l.210: 'which is a collection of all the moorings': Is this database only composed of moorings? To my knowledge this database is also composed of repeated ship CTD transects.

l.250: in the datasets, instead of correcting the already available data, I think it will be better to flag those data.

l.299: How is a 4km resolution ice cover representative of the sea ice cover in a fjord? Is this resolution good enough for the size of a fjord?

l.301: in addition to the sea ice cover, it will be of great interest to get the sea ice thickness. Combined with the sea ice cover, this will give an estimate of the sea ice volume, the real indicator of the sea ice loss in the Arctic. Does this dataset gather this information too?

l.335: why is the unit of the nutrients umol/L? umol/kg will be consistent with the units of TA and DIC.

l.337: I am surprised that there are no data for the biology drivers in Storfjorden. There has been several cruises in the region that collected biology drivers. How do you decide which data are merged in your dataset?

Table 3: what is Q?

Figure 5: I am not sure about the interest of this figure. There is indeed correlation in between those variables, but this does not imply causality, and there is no explanation of the correlation between the variables in the manuscript.

l.419: I don't really see the point of section5, as it just shows some correlations but does not explain them really.

---

## Author Comment (AC1)

**Reviewer 1**

We thank the reviewer for taking the time to provide a useful and considered review of this manuscript, and the dataset detailed therein. We have reflected on the comments, made changes to the manuscript where appropriate, and provide below a point-by-point reply. The dataset itself is already published on PANGAEA, but all of the comments made here will be integrated into the development of v2.0, which will be uploaded to PANGAEA early next year. That being said, a live version of the dataset will be directly benefiting from your comments, and is available via a user interface that may be accessed via the WP1 website by clicking the 'Data access' tab.

We have addressed specific comments in line with the reviewers original text.

This is a very good step to provide an access to the coastal Arctic data.

It is worth pointing out here that the focus of the dataset is to provide data within seven specific Arctic fjords, not for coastal or open waters.

However some data are easy to obtained and self-explanatory (SST, ice cover, glaciers balance etc) some data are difficult to obtain, yet are simple (fisheries landings, jobs, tourists movements etc) and some data need to be carefully presented, as they have very broad meaning - in this work are data "species richness".

We agree with the reviewer that the data contained in this amalgamation range in difficulty and utility.

This for biologist means nothing. Species richnes in what taxon ? (there are no in the world complete species lists for the one site). It can be the species richness of mesozooplankton from specific water column (say 0-200 m) or microplankton species from euphotic waters (in spring ? in summer ?, all year ? ) - all taxa of microplankton including Ciliata and Amoeba and other little known groups ? Same questions go for all groups of benthos (what habitat is presented ? rocky shore or soft bottom near the glaciers ? etc)

We agree that the 'species richness' field provided in this amalgamated dataset does not quite provide the intended information for which it was created. As detailed in the manuscript, the types of data provided in this dataset were determined via a published review paper on key drivers of change in EU Arctic fjords. Because species richness data tended to be lacking, we made a first attempt to fill this gap. v2.0 of the dataset will incorporate your feedback to provide a more specific accounting of the richness of species by different functional groups and/or taxa.

As to the data completeness there is much more to dig out - probably the best is to use the research institutions with long term observations (like authors have identified Norwegian Danish and German projects). There is also the Institute of Geophysics Polish Academy of Sciences research station in Hornsund, Svalbard with meteo data complete from 1957 to today, and glaciers balance last 40 years, oceanographic (SST) last 15 years.

The search for new open source datasets, that are without sharing restrictions, is an ongoing process that will continue for the length of the FACE-IT project, for which this amalgamation of data was commissioned. We thank the reviewer for their suggestions on these additional sources to investigate. Specifically though, Hornsund falls outside of the FACE-IT study sites, which is why no data were collected for it. As indicated in Section *2.3.5. (Additional sources of note)*, meteorological data also fall outside of the scope of this dataset

There is Institute of Oceanology PAS with r/v OCEANIA sailing every summer to the Svalbard coastal waters since 1989, with complete data set on hydrology, mesozooplankton and benthic life of Hornsund and Kongsfjorden.

We thank the reviewer for their suggestion on investigating the use of data from the r/v OCEANIA. While open ocean data is outside of the scope of this dataset, data within Kongsfjorden is very much something we want to include. Querying the dataset, we see that it does contain 8,869 OCEANIA hydrographic data points (temperature and salinity) from 1997 to 2003 (via PANGAEA) within both Kongsfjorden and Isfjorden. Investigating this issue more closely, we see that it also contains the zooplankton datasets as well. We will ensure in v2.0 that all available applicable data from the OCEANIA cruises are included.

There are Norwegian long time benthic-photo surveys for over 40 years in the rocky sublittoral of several Svalbard fjords (University of Tromso and AKVAPLAN NIVA) .

Photos surveys (unless converted to a spreadsheet of presence absence etc.) were determined to be outside of the scope of this dataset. The reason for this is that all of the data collected, from glaciology to national statistics, need to be able to be presented together in a single spreadsheet or text file. Since photographs cannot be included in such files, it has not been possible to include them in this dataset. The reasoning for this omission of photo survey data has been added to the introduction of the revised manuscript.

Lots of primary production data from Svalbard fjords have been published, need to be digged out from papers.

While primary production data are lacking in this amalgamated dataset, and are a top priority for v2.0, the extraction of data points from published literature is a high cost as well as a low accuracy and return activity that is avoided in favour of locating sources of FAIR data. It is for this reason that there is so much effort in the development of the FAIR data principles. In v1.0 of the dataset, many individual PAR data points for Kongsfjorden were harvested from published literature. It is from this experience that this method of collecting data is avoided.

I understand that authors used available data repositories, unfortunately these are still holding only a fraction of known information about coastal Arctic. The challenge is very difficult, and presented paper is a nice step, but only a minor step.

We agree with the reviewer that the amount of data not accounted for here is greater than the data that has been compiled. This is indeed a challenge in science, and one that must be addressed by the scientific community. Much work is being done in this regard, and many teams are also digitising pre-digital data in order to make them FAIR. It is regrettable that some data are sitting in drawers or in non-accessible databases and repositories. The work continues and the documentation for the assembly of this dataset is a step forward in that process. We are glad that you acknowledge that our efforts are a valuable step but beg to differ with the qualifier "minor". It has taken roughly one year to compile v1.0 of the dataset.

---

## Author Comment (AC2)

**Reviewer 2**

Review of the manuscript: "A dataset for investigating socio-ecological changed in Arctic Fjords."

This work manuscript presents the compilation of various data from Arctic Fjords, to investigate the socio-ecological changes in 7 Arctic Fjords. The manuscript is very clear and well written, the dataset is well presented and useful for several type of studies. However, this dataset is just a compilation of datasets already available in several databases, although I think that much more data are available but maybe not publicly. I suggest publishing it after major revisions. Please find my comments beneath.

We thank the reviewer for taking the time to provide a considered and detailed review of both the manuscript and the dataset it documents. All comments have been considered at length and acted upon accordingly. Specific comments have been addressed below and changes to the manuscript have been made. While the dataset itself is already published on PANGAEA, all of the reviewers comments w.r.t. data have been documented and will be implemented in v2.0 of the dataset, which will be submitted for publication early next year. That being said, a live version of the dataset will be directly benefiting from your comments, and is available via a user interface that may be accessed via the WP1 website by clicking the 'Data access' tab.

Main comment:

My main comment is about the added value of this dataset. I understand that this is a compilation of already publicly available datasets. Although of interest for some studies I think such a compilation of data should aim for more than 7 fjords in the arctic, and rather try to compile all available data for Greenland and Svalbard fjords for example. A lot of data is available in a lot of fjords in Greenland, but maybe not easily available indeed. However, considering those data will add a lot of value to this dataset, that could be used to assess for example the impact of climate change on the Arctic Fjords.

We agree with the reviewer that many more sites and datasets for them could be added to this amalgamation of published data. We think the reviewer would agree that it is necessary to define a specific scope for the data collection, as there is simply too much data for the wider Arctic to amalgamate. All of the fjords for Greenland and Svalbard is certainly an ideal target, but we argue that for the purpose of the investigation of socioecological changes, one must select sites with data for as many of the 14 key drivers as possible. We also posit that at some point there would be rapidly diminishing returns on the effort put into the amalgamation. For example, while it would require months to be confident that all FAIR data were amalgamated for a given site, it would not

be particularly helpful to socioecological investigations to include sites which have primarily hydrographic data (e.g. Rijpfjorden), but are lacking data for most of the other categories (e.g. biological, social, etc.).

Even with this consideration, there are very few EU Arctic fjords with the full range of key drivers sampled. It was therefore established as another primary criterion that a range of fjords which might best represent the different stages of the changing Arctic climate be chosen. For example, Young Sound is not yet significantly affected by climate change, unlike Kongsfjorden and Isfjorden, which are rapidly borealising. Porsangerfjorden is now relatively ice free. Each of the seven chosen fjords has a reason why they were selected, and when taken together we think they cover a valid and interesting scientific range of investigations, while still remaining feasible for a small team of two people to manage. This is why we settled on these seven fjords and why we think this dataset should remain limited to them. We have added this more detailed explanation to the Intro section.

Minor comments:

How do the authors define the Arctic? Is it the Arctic circle? I am a bit surprised that a fjord in Northern Norway is included in the manuscript. Please clarify.

We thank the reviewer for pointing out this oversight in the manuscript. The range for the EU Arctic used here is taken from the Copernicus definition, which closely matches the AMAP definition of the Arctic circle. This has now been stated explicitly in the Intro section.

L30: 'range of habitats for many important species': do you have any reference for that?

The references for the following sentence are meant to support the statement made in this sentence. We have reduced the strength of the statement, and combined the two sentences to more clearly cite the reference.

L34: Please delete 'extreme'

Agreed, replaced with 'polar'.

L37: This sentence argues that most of the sampling in the Arctic is performed by large cruise ships. I think it should be clarified here that you are talking about the coastal Arctic and I guess about surface data. In my knowledge, the central Arctic Ocean is mainly occupied by research ship so far, and most of the data come from autonomous platforms.

We thank the author for pointing out this inconsistency in the text. The use of the expression 'cruise ship' was an error. We meant to say 'research ship', and the text has been corrected accordingly. We have also updated the sentence to mention the use of autonomous platforms for data collection.

l.134: The manuscript explained the source of the data and the type of data, but for the ocean temperature for example, how are the data organized? Are they daily averaged or is it just a compilation of all available data in the fjord? For example, in Kongfjorden, several temperature datasets are available, moorings but also CTD from ships for example... Is that indicated in the dataset?

Thank you for highlighting the lack of clarity on this point. While it is mentioned in the caption of Figure 2 that the counts shown are for daily data points, it is not detailed explicitly in the text. We have therefore added two sentences to the end of Section 2.2 to make it clear that all data in this dataset are either daily, monthly, or annual. And that if data were available at a finer resolution, daily averages were created.

L.183: Did the author look at the arctic data center for more datasets? Is it on purpose that the fjords are only located on the Eurasian side of the Arctic. I could expect that quite a few data are also available in the Canadian Archipelago.

While none of the initial datasets required for this project, as detailed in the Intro section, were hosted on the Arctic Data Centre (ADC), this repository was identified and flagged as useful during the broader data amalgamation process. The ADC contains very few datasets within the 7 study sites outlined for this work with data that fall within the 14 drivers of interest, and of those that do (e.g. moorings) were already sourced via the sources detailed in Section 2.3. That being said, we will continue to monitor the datasets available on the ADC and ensure that during the creation of v2.0 of the dataset that all relevant data are included.

l.210: 'The first of these is the UNIS database': please add a reference

Added in text reference and citation:
Skogseth, R., Ellingsen, P., Berge, J., Cottier, F., Falk-Petersen, S., Ivanov, B., … Vader, A. (2019). UNIS hydrographic database [Data set]. Norwegian Polar Institute. https://doi.org/10.21334/unis-hydrography

l.210: 'which is a collection of all the moorings': Is this database only composed of moorings? To my knowledge this database is also composed of repeated ship CTD transects.

The reviewer is correct, the database is composed of ship transects, in addition to mooring data. We have corrected the text accordingly. Specifically, the UNIS database aims to provide users with temperature+salinity profiles.

l.250: in the datasets, instead of correcting the already available data, I think it will be better to flag those data.

This is an important point, and one we discussed at length before submitting the manuscript. We agree that it is outside of the scope of a project like this to correct values in already published data however; it is anticipated that most users of this dataset will not consider any flags on the data, but will rather use the data 'as is'. Indeed, there are so many different types of data amalgamated together in this dataset that constructing a consistent unified flagging scheme would itself be a new project. Looking back over our records, it appears that in the end we did not perform any systematic corrections on any published data. Opting instead to not include the problematic datasets. The text has been corrected to make it clear to the reader that rather than flagging data, dubious data points or entire datasets are rather removed from the amalgamation.

l.299: How is a 4 km resolution ice cover representative of the sea ice cover in a fjord? Is this resolution good enough for the size of a fjord?

One may see the resolution of the 4 km ice cover product within the fjords via the pop-out windows in Figure 1. The use of these data within Young Sound, and perhaps Kongsfjorden, is dubious, but it is certainly high enough resolution for the other 5 study sites. It is however necessary to have a consistent product across all 7 sites, with as long of a time series as possible, which is why it was finally decided to use the ice cover product in question. A comparison was made to the 1 km product, which has a much shorter time series, and the values correlated very closely. In preference, therefore, of the longer time series, the 4 km product was chosen. The drawback of using this dataset is explicitly mentioned in the revised manuscript.

l.301: in addition to the sea ice cover, it will be of great interest to get the sea ice thickness. Combined with the sea ice cover, this will give an estimate of the sea ice volume, the real indicator of the sea ice loss in the Arctic. Does this dataset gather this information too?

We agree with the reviewer that the knowledge of sea ice thickness would be ideal, but to our knowledge no such product exists. We learned during the creation of this amalgamated dataset that there is no consistent *in situ* measuring of sea ice cover and/or thickness throughout the Arctic. The MASIE 4 km sea ice extent product is the longest running complete measurement of sea ice. It is unfortunate that it lacks a thickness data layer. The revised manuscript indicates this gap.

l.335: why is the unit of the nutrients umol/L? umol/kg will be consistent with the units of TA and DIC.

The difference between the two is due to the necessity for the precision of the density of seawater when calculating various aspects of seawater carbonate chemistry. Because the salinity and temperature of seawater can have an effect on the calculation of TA etc., these values are generally given as µmol/kg. For nutrients however this is not the case. So the simpler to measure umol/l is used. Also considering that most nutrient sampling is done in tandem with other biologically oriented water samples, which tend to be measured in units of litres, rather than kilograms.

It is also an issue that one needs temperature+salinity to convert from litres to kg, but one doesn't always have these values with biological data, so the conversion is not possible, even though that is the best practice. This explanation (and reference) has been added to the text.
Jiang L-Q, Pierrot D, Wanninkhof R, Feely RA, Tilbrook B, Alin S, Barbero L, Byrne RH, Carter BR, Dickson AG, Gattuso J-P, Greeley D, Hoppema M, Humphreys MP, Karstensen J, Lange N, Lauvset SK, Lewis ER, Olsen A, Pérez FF, Sabine C, Sharp JD, Tanhua T, Trull TW, Velo A, Allegra AJ, Barker P, Burger E, Cai W-J, Chen C-TA, Cross J, Garcia H, Hernandez-Ayon JM, Hu X, Kozyr A, Langdon C, Lee K, Salisbury J, Wang ZA and Xue L (2022) Best Practice Data Standards for Discrete Chemical Oceanographic Observations. Front. Mar. Sci. 8:705638. doi: 10.3389/fmars.2021.705638

l.337: I am surprised that there are no data for the biology drivers in Storfjorden. There has been several cruises in the region that collected biology drivers. How do you decide which data are merged in your dataset?

We agree with the reviewer that it is unfortunate that there is a lack of biology data for Storfjorden. We are aware that there are biological data being collected, but to our knowledge and at the time of this writing, these data are not publicly available on any of the data portals we use. Decision on what to merge in the dataset is first guided by the 14 key drivers of change in EU Arctic fjords, then by the data found at the primary data portals. Data that are not FAIR are not able to be merged into this dataset.

Table 3: what is Q?

Q is a measurement of flow rate (m^3 s-1). It is used primarily to measure river flow rates, but is also used for glacial runoff. I have made a note of this in the caption for table 3.

Figure 5: I am not sure about the interest of this figure. There is indeed correlation in between those variables, but this does not imply causality, and there is no explanation of the correlation between the variables in the manuscript.

We agree, this figure has been replaced with one showing an analysis that can more directly explain a causal relationship between two variables.

l.419: I don't really see the point of Section 5, as it just shows some correlations but does not explain them really.

We agree with the reviewer and have removed this section from the main body of the text. Due to the internal variability of this model forecast dataset for the FACE-IT consortium it was deemed desirable to use it in a publication. But considering that these data are not actually included within the data amalgamation being documented here, it is extraneous to the manuscript.

That being said, these results may be of interest to researchers working in these specific fjords, so this section has been moved to the supplementary. We have also included a short paragraph to the discussion that summarises these findings to the reader/user.